# Robust Offline Imitation Learning from Diverse Auxiliary Data

**Udita Ghosh**                                              *ughos002@ucr.edu*
*University of California, Riverside*

**Dripta S. Raychaudhuri**                                   *drayc001@ucr.edu*
*AWS AI Labs* *

**Jiachen Li**                                               *jiachen.li@ucr.edu*
*University of California, Riverside*

**Konstantinos Karydis**                                     *kkarydis@ece.ucr.edu*
*University of California, Riverside*

**Amit K. Roy-Chowdhury**                                    *amitrc@ece.ucr.edu*
*University of California, Riverside*

**Reviewed on OpenReview:** *https://openreview.net/forum?id=Hy2KAldqAo*

## Abstract

Offline imitation learning enables learning a policy solely from a set of expert demonstrations, without any environment interaction. To alleviate the issue of distribution shift arising due to the small amount of expert data, recent works incorporate large numbers of auxiliary demonstrations alongside the expert data. However, the performance of these approaches rely on assumptions about the quality and composition of the auxiliary data, and they are rarely successful when those assumptions do not hold. To address this limitation, we propose *Robust Offline Imitation from Diverse Auxiliary Data* (ROIDA). ROIDA first identifies high-quality transitions from the entire auxiliary dataset using a learned reward function. These high-reward samples are combined with the expert demonstrations for weighted behavioral cloning. For lower-quality samples, ROIDA applies temporal difference learning to steer the policy towards high-reward states, improving long-term returns. This two-pronged approach enables our framework to effectively leverage both high and low-quality data without any assumptions. Extensive experiments validate that ROIDA achieves robust and consistent performance across multiple auxiliary datasets with diverse ratios of expert and non-expert demonstrations. ROIDA effectively leverages unlabeled auxiliary data, outperforming prior methods reliant on specific data assumptions. Our code is available at https://github.com/uditaghosh/roida.

## 1 Introduction

Integration of deep neural networks in reinforcement learning (RL), coupled with the development of efficient training algorithms, has yielded remarkable performance across a wide variety of sequential decision-making tasks, such as playing games (Mnih et al., 2015; Silver et al., 2017; 2018) and solving complex robotics tasks (Duan et al., 2016; Levine et al., 2016; Kaufmann et al., 2023). Despite this progress, two challenges still remain: the need for extensive environment interactions (Levine et al., 2020), and the inherent difficulty in designing reward functions for complex real-world tasks (Abbeel & Ng, 2004).

Imitation learning (IL), where an agent learns directly from task demonstrations, has been employed as one way to tackle the aforementioned challenges (Abbeel & Ng, 2004; Ross & Bagnell, 2010; Ho & Ermon, 2016).

---

*Work done outside Amazon.

IL methods can be categorized as online or offline. Online IL algorithms rely on experiences gathered from the environment by executing intermediate policies during training (Ho & Ermon, 2016; Kostrikov et al., 2019). However, online interaction may be infeasible, unsafe, or expensive in many real-world settings. Offline IL provides a safer alternative, where agents learn solely from pre-collected expert demonstrations without environmental interaction. Offline IL methods like behavioral cloning (BC) (Bojarski et al., 2016) remove the need for online experience. However, offline imitation remains vulnerable to distribution shifts as a result of error accumulation over time (Ross et al., 2011).

To address the challenge of distribution shift, recent offline IL methods incorporate a substantial number of auxiliary imperfect demonstrations alongside expert demonstrations. These auxiliary demonstrations are not expected to meet any optimality criteria, encompassing a mix of expert, near-expert, and non-expert trajectories. A recent work, DWBC (Xu et al., 2022), treats this auxiliary data as a mixture of expert and sub-optimal data, and trains a discriminator for weighted behavioral cloning. On the other hand, DemoDICE (Kim et al., 2022) performs state-action distribution matching on the auxiliary data as a regularization term, in addition to matching the distribution over the expert set by solving a convex optimization problem in the dual space. Both methods share the assumption that some high-reward behavioral data are present in the auxiliary dataset, and consequently, utilize only these expert transitions for policy learning by filtering out the non-expert trajectories. However, as shown in Fig. 1, their performance fluctuates as the proportion of expert data in the auxiliary set varies, since they fail to leverage the information available in the non-expert data. Although non-expert data may not explicitly provide knowledge of the optimal policy, it contains substantial dynamics information for the agent. In practical scenarios, it is highly unlikely that the quality of the demonstration data in the auxiliary dataset will be known *a priori*. Thus, we design an offline IL algorithm that is more robust to the demonstration quality in the auxiliary data. Our approach does not make any assumptions about the quality of the auxiliary data and achieves reasonably consistent performance as the proportion of high quality to low quality data varies, as shown in Fig. 1.

In this paper, we present Robust Offline Imitation from Diverse Auxiliary Data (ROIDA), an algorithm that combines the simplicity of BC with the capability of offline RL to leverage transition data of varying quality in the auxiliary dataset. ROIDA does not impose any assumptions on the composition of the auxiliary dataset, and can effectively utilize diverse auxiliary datasets encompassing different ratios of expert and non-expert demonstrations. To achieve this, we identify potential expert state-action pairs in the auxiliary set and assign large weights to these samples in the subsequent weighted BC objective. This involves two key steps: 1) training a discriminator to distinguish between expert and non-expert transitions using Positive-Unlabeled (PU) learning (Elkan & Noto, 2008), and 2) applying weighted BC to all state-action pairs in the auxiliary data, with weights derived from importance sampling ratios based on thresholded scores provided by the discriminator. However, as previously mentioned, the auxiliary dataset may lack expert state-action pairs. To address sub-optimal transitions, we perform temporal difference learning using the importance sampling ratios from the discriminator as rewards. This approach aims to guide the policy toward the expert

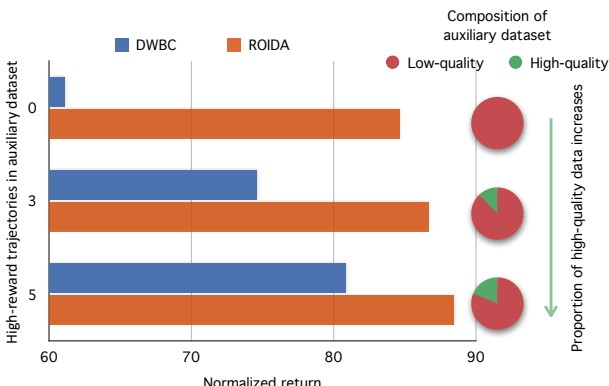

Figure 1: **Robustness to composition of auxiliary data.** Performance of existing offline IL algorithms, such as DWBC (Xu et al., 2022), varies significantly depending on the amount of high-quality transitions present in the auxiliary data (given expert set is kept fixed). In contrast, ROIDA is more robust, highlighting its ability to extract information even from low-quality transitions. The setup shown here is on the Hopper environment; refer to Sec. 5 for details.

states, thereby improving long-term returns (as measured by the discriminator) on states not observed by experts. This guidance allows ROIDA to extract value from low-quality transitions, in addition to expert behaviors, in contrast to previous works.

Experiments on the D4RL benchmark (Fu et al., 2020) show that ROIDA consistently achieves high performance across seven environments, using auxiliary datasets with varying proportions of expert data. This consistent success highlights ROIDA's ability to leverage diverse unlabeled data without assumptions on data quality. In contrast, existing offline IL methods perform well only in selective scenarios that match their specific assumptions about the data composition. *Our approach is the first to relax the data quality assumptions of the auxiliary dataset, utilizing it to gain comprehensive knowledge about the expert policy and the environment.* No other method fully utilizes both the expert and sub-optimal demonstrations in the auxiliary data for policy learning, leading to suboptimal performance.

To summarize, our primary contributions are as follows:

1. We analyze state-of-the-art (SOTA) offline IL methods that utilize auxiliary data along with a small expert set. Our empirical analysis highlights the unrealistic assumptions of these methods, particularly regarding the composition of the auxiliary set. With this in mind, we design an offline IL algorithm, ROIDA, that addresses the limitations of these different methods and remains robust to the quality of demonstrations in the auxiliary dataset.

2. ROIDA incorporates PU learning alongside temporal difference learning to effectively utilize both expert and sub-optimal transitions in the auxiliary data.

3. We empirically validate that ROIDA is robust to the quality of the auxiliary data and consistently achieves high performance across different environments.

## 2 Related Works

### 2.1 Imitation learning

Imitation learning (Schaal, 1999) leverages expert demonstrations to train a policy that successfully mimics the expert's behavior. A common approach is behavioral cloning (BC) (Pomerleau, 1989; Bojarski et al., 2016), which frames policy learning as a supervised learning problem. However, BC exhibits sub-optimal performance in states distant from the training data (Ross et al., 2011). Alternatively, inverse reinforcement learning (IRL) first learns a reward function to explain the demonstrated actions before using it to train a policy through any RL algorithm. While popular IRL algorithms (Ho & Ermon, 2016; Fu et al., 2017; Abbeel & Ng, 2004; Ziebart et al., 2008) can outperform BC, the majority are online methods requiring a substantial number of environment interactions during training, resulting in poor sample efficiency. To circumvent the need for environment interactions, several offline IRL methods (Kostrikov et al., 2019; Swamy et al., 2021; Garg et al., 2021) based on adversarial training have been proposed. However, these approaches assume that all demonstrations are equally good, resulting in performance degradation when demonstrations contain sub-optimal data, as in our case.

### 2.2 Learning from noisy demonstrations

Various approaches have been introduced to address the challenge of imitation learning from sub-optimal or noisy experts (Wu et al., 2019; Tangkaratt et al., 2020; Brown et al., 2019; Wang et al., 2021; Sasaki & Yamashina, 2021). However, many of these works rely on strong assumptions about the dataset, such as the expert data dominating the majority of the offline dataset (Sasaki & Yamashina, 2021) or defining sub-optimality as additive Gaussian noise to the action (Tangkaratt et al., 2020). Other works assume that trajectories are provided with labels indicating degree of optimality (Wu et al., 2019; Wang et al., 2021) or preference rankings between trajectory pairs (Brown et al., 2019). Furthermore, these approaches require environment interactions during learning while we focus on the offline setting.

The offline IL setup with an auxiliary dataset was first explored in DemoDICE (Kim et al., 2022). DemoDICE conducts state-action distribution matching over the expert set and introduces a regularization constraint to ensure the learned policy remains close to the behavior policy of the auxiliary dataset. DWBC (Xu et al., 2022) treats the auxiliary data as a mixture of expert and sub-optimal data, and utilizes positive-unlabeled learning to train a discriminator for weighted behavioral cloning. Both approaches encounter challenges

when the auxiliary data is highly sub-optimal and might even exhibit inferior performance compared to counterparts utilizing only expert data. In a recent work (Shao et al., 2023), the authors propose an offline IL algorithm specifically for cases where no expert data is present in the auxiliary dataset. This approach assigns a reward of 1 to the expert transitions and 0 to all auxiliary transitions, employing an offline RL approach alongside BC. While this design can enhance performance when there is no expert data in the auxiliary dataset, the assignment of zero reward to all auxiliary transitions can lead to poor performance when the proportion of experts in the auxiliary dataset is increased. In practical settings, it is highly unlikely that the data quality in the auxiliary dataset will be known beforehand. With this in mind, we design an offline IL algorithm that addresses the limitations of these different approaches and remains robust to the quality of demonstrations in the auxiliary dataset.

### 2.3 Offline reinforcement learning

Offline RL (Levine et al., 2020) aims to learn policies by utilizing static offline datasets without requiring additional interactions with the environment. Notably, in offline RL, the training dataset is permitted to contain non-optimal trajectories, and the reward for each state-action-next state transition triplet is known. Our algorithm takes inspiration from a subset of methods within the offline RL literature, specifically those employing filtered advantage weighted regression (Peng et al., 2019; Wang et al., 2020; Nair et al., 2020) and behavioral cloning augmented off-policy learning (Fujimoto & Gu, 2021) In another recent work, UDS (Yu et al., 2022) utilizes an auxiliary set without reward labels in addition to the usual reward labeled offline RL dataset. The approach applies zero rewards uniformly to any unlabeled data and can be effective in highly specific offline RL scenarios. Different from this related work, we do not have access to any reward-labeled dataset in the offline IL setting.

## 3 Problem Setting

We formulate our problem using the standard fully-observable Markov Decision Process (MDP) framework (Sutton & Barto, 2018). An MDP $\mathcal{M}$ is characterized by the tuple $(\mathcal{S}, \mathcal{A}, \mathcal{T}, r, \gamma, p_0)$, where $\mathcal{S}$ denotes the state space, $p_0$ denotes the initial state distribution, and $\mathcal{A}$ represents the action space. At each time step $t$, given a state $s_t \in \mathcal{S}$, the agent selects an action $a_t \in \mathcal{A}$ according to its policy $\pi(a_t|s_t) \in \Delta(\mathcal{A})$, where $\Delta(\mathcal{A})$ denotes the probability simplex over $\mathcal{A}$. Following the execution of action $a_t$, the MDP transitions to a new state $s_{t+1} \in \mathcal{S}$ based on the transition probability $\mathcal{T}(s_{t+1}|s_t, a_t)$, while the agent receives a reward $r(s_t, a_t) \in \mathbb{R}$. The primary objective for the agent is to maximize the expected discounted reward $\mathbb{E}\left[\sum_t \gamma^t r(s_t, a_t)\right]$ with discount factor $\gamma \in [0, 1]$. The state-action distribution of this policy $\pi$ under the transition function $\mathcal{T}$ is defined as $d^\pi = (1 - \gamma) \sum_t \gamma^t d_t^\pi$, where $d_t^\pi$ is the distribution of $(s_t, a_t)$ under $\pi$ at step $t$.

In the offline IL setup we do not have access to the reward function $r$. Instead, we utilize a set of demonstrations provided by an expert policy $\pi_E$ in the form of a dataset of expert tuples $\mathcal{D}_E = \{(s_i, a_i, s_i')\}_{i=0}^{N_E}$, where $(s, a)$ is sampled from $d^{\pi_E}$ and $s'$ is sampled from $\mathcal{T}(s'|s, a)$. Additionally, we assume access to a substantial amount of pre-collected demonstrations $\mathcal{D}_O = \{(s_i, a_i, s_i')\}_{i=0}^{N_O}$ gathered by some unknown behavior policy from the distribution $d^{\pi_O}$ ($N_O \gg N_E$). It is important to note that these tuples are not presumed to satisfy any optimality criteria for the specific task at hand. Given this *expert set* $\mathcal{D}_E$ and the *auxiliary set* $\mathcal{D}_O$, our objective is to learn a policy $\pi^*$ capable of maximizing the unknown reward $r$, without the need for direct interaction with the environment.

## 4 Method

### 4.1 Overview

As discussed in Sec. 3, the auxiliary demonstrations are not expected to adhere to any optimality criteria, and can contain a mix of expert, near-expert, and non-expert trajectories. Existing methods rely on certain assumptions about the quality of this data to effectively utilize them. DWBC (Xu et al., 2022) and DemoDICE (Kim et al., 2022) assume the presence of high-reward expert data in the auxiliary dataset. In

practice, knowing the data quality in the auxiliary dataset beforehand is highly unlikely. Thus, we propose ROIDA to effectively leverage transition data of varying quality in the auxiliary dataset. ROIDA uses two key ideas to achieve this.

*First*, ROIDA aims to emulate expert behavior by considering both expert demonstrations and any task-optimal state-action pairs present in the auxiliary dataset. To accomplish this, we employ a discriminator $d(s, a)$ trained using PU learning to approximate a reward $\tilde{r}(s, a)$ for each state-action tuple in the auxiliary set (Sec. 4.2). This reward assesses the optimality of each data point in the auxiliary set compared to the expert set. When the reward surpasses a designated threshold, ROIDA includes the corresponding instance as an approximate expert data point, applying weighted BC on this sample with the discriminator output acting as the weight (Sec. 4.3).

However, the auxiliary dataset may contain a significant number of state-action pairs that do not exceed this reward threshold. Rather than completely excluding these samples from optimization, ROIDA incorporates its *second* critical element: leveraging the transition information in the data via temporal difference learning, using the estimated rewards (Sec. 4.4). This approach aims to steer the policy toward high-reward states, thereby enhancing long-term returns on states for which optimal actions are not readily available. This two-pronged guidance enables ROIDA to extract value from low-quality transitions in addition to expert behavior, without imposing any assumptions on auxiliary dataset composition. Fig. 2 presents an overview of our approach.

## 4.2 Learning a reward model

In order to perform both weighted BC and temporal difference learning on the auxiliary dataset, we construct a reward model by training a discriminator $d(s, a)$ to discern between expert and sub-optimal transitions. Unlike prior approaches (Ho & Ermon, 2016; Kim et al., 2022) that treat this as a standard binary classification task, designating all samples from $\mathcal{D}_O$ as negative, we opt for PU learning in the discriminator training process. This decision is driven by the potential presence of expert transitions within the auxiliary dataset. As a result, we consider the auxiliary dataset as an unlabeled set, encompassing both positive samples (expert state-action transitions) and varied negative samples (non-expert transitions), with the expert dataset serving as the labeled positive dataset.

Figure 2: **Framework overview.** ROIDA first learns a reward function using PU learning. It then identifies high-reward expert-like transitions and combines them with the expert data for weighted BC (Sec. 4.2,4.3). To extract value from lower quality samples, ROIDA applies TD learning, steering the policy towards high reward states (Sec. 4.4). By combining weighted BC and TD learning, ROIDA effectively leverages uncurated offline data.

The core idea in PU learning is to re-weight the different losses for the positive and the unlabeled data in an effort to derive an estimate of the model loss on negative samples, which is not directly accessible. Due to the limited amount of expert data, we use a non-negative risk estimator described in (Kiryo et al., 2017) in order to make the discriminator more robust:

$$\min_d \ \eta \ \mathbb{E}_{(s,a)\sim\mathcal{D}_E} \left[ -\log d(s, a) \right] + \max\left( 0, \ \mathbb{E}_{(s,a)\sim\mathcal{D}_O} \left[ -\log(1 - d(s, a)) \right] - \eta \ \mathbb{E}_{(s,a)\sim\mathcal{D}_E} \left[ -\log(1 - d(s, a)) \right] \right) . \quad (1)$$

Here, $\eta$ is a hyperparameter that represents the positive class prior.

Given the trained discriminator, we calculate the reward for each state-action tuple in the auxiliary dataset as follows,

$$\tilde{r}(s, a) = \log \frac{d(s, a)}{1 - d(s, a)} \ , \tag{2}$$

where $d(s, a)$ is clipped to the range of $[0.1, 0.9]$ to prevent unbounded rewards. All samples in $\mathcal{D}_E$ are assigned a value of $d(s, a) = 0.9$ and the reward is calculated accordingly.

The specific form of the reward is inspired by DICE (DIstribution Correction Estimation) approaches (Kim et al., 2022; Ma et al., 2022) which try to estimate $\log \frac{d^{\pi_E}(s, a)}{d^{\pi_O}(s, a)}$. This ratio serves as an indicator for the importance of a state-action tuple; higher values mean that the expert often takes the action $a$ at state $s$. While DICE methods perform this estimation by training a simple binary classifier, we use PU learning to train a more robust discriminator to prevent treating expert samples from $\mathcal{D}_O$ as negatives.

### 4.3 Reward-weighted behavioral cloning

Using the rewards obtained from the discriminator, we can identify the expert transitions in the auxiliary dataset for policy training. Instead of directly employing the rewards as weights, we employ a direct thresholding scheme to exclude highly sub-optimal state-action tuples. Finally, we use the filtered samples, alongside those from the expert dataset, to perform weighted behavioral cloning:

$$\min_{\pi} \mathop{\mathbb{E}}_{(s,a) \sim \mathcal{D}_E} \left[ -\log \pi(a|s) \right] + \alpha \mathop{\mathbb{E}}_{(s,a) \sim \mathcal{D}_O} \left[ -\log \pi(a|s) \cdot \tilde{r}(s, a) \cdot \mathbb{1}[\tilde{r} > \tau] \right] \ . \tag{3}$$

Here, $\tau$ is a hyperparameter that governs the strength of thresholding and helps balance between excluding sub-optimal transitions and incorporating expert transitions from the auxiliary dataset. Hyperparameter $\alpha$ is used to weigh in the overall BC loss from the auxiliary dataset. The auxiliary dataset may include state-action pairs which fall below the reward threshold. Instead of discarding them, we integrate these samples into the training process via temporal difference (TD) learning, as elaborated below.

### 4.4 TD learning using learned rewards

Despite not meeting the reward threshold, sub-optimal state-action tuples possess the potential to enhance the learned policy, augmenting its robustness against distribution shifts during deployment. This is due to the broader coverage of the state space within the auxiliary dataset, surpassing the limited span of the small expert dataset. In the absence of access to expert behavior in these states for direct learning, we employ a *shortest path* strategy to leverage these samples. More precisely, our objective on these states is to steer the policy toward states observed by the expert and subsequently imitate the expert's behavior accordingly. To accomplish this, we use a TD3-style (Fujimoto & Gu, 2021) algorithm to learn a Q-function using the approximated rewards and then direct the policy to maximize the long-term return on these expert-unobserved states. The Q-function is learned as follows,

$$\mathop{\arg\min}_{Q} \sum_{(s,a,s') \sim \mathcal{D}_E \cup \mathcal{D}_O} \|\mathcal{B}^{\pi} Q(s, a) - Q(s, a)\|^2 \ , \tag{4}$$

where $\mathcal{B}^{\pi}$ denotes the Bellman operator, that is

$$\mathcal{B}^{\pi} Q(s, a) = \tilde{r}(s, a) + \gamma \sum_{a' \in A} \left[ \pi(a'|s') Q(s', a') \right] \ . \tag{5}$$

Using this Q-function we can formulate our refined policy learning objective as

$$\min_{\pi} \mathop{\mathbb{E}}_{(s,a) \sim \mathcal{D}_E} \left[ -\log \pi(a|s) \right] + \alpha \mathop{\mathbb{E}}_{(s,a) \sim \mathcal{D}_O} \left[ -\log \pi(a|s) \cdot \tilde{r}(s, a) \cdot \mathbb{1}[\tilde{r} > \tau] \right] + \beta \mathop{\mathbb{E}}_{s \sim \mathcal{D}_E \cup \mathcal{D}_O} \left[ -Q(s, \pi(s)) \right] \ . \tag{6}$$

Here, $\beta$ is a hyperparameter which controls the contribution of the Q-loss. By maximizing this Q-function alongside the weighted BC objective, we incentivize the policy to: 1) act optimally in states where we have

expert actions, and 2) guide the agent efficiently from expert-unobserved states to expert-observed states and act optimally subsequently.

The pseudo-code for the overall framework is presented in Algorithm 1.

---

**Algorithm 1** Robust Offline Imitation from Diverse Auxiliary Data (ROIDA )

---

**Require:** Dataset $\mathcal{D}_E$ and $\mathcal{D}_E$, hyperparameter $\eta, \alpha, \beta, \gamma$
 1: Initialize the imitation policy $\pi$, the discriminator $d$ and Q-function approximator $Q$
 2: Train discriminator $d$ with non-negative PU learning following Eqn. 1
 3: **for** t=1 to T **do**
 4:    Sample $(s_e, a_e) \sim \mathcal{D}_E$ and $(s_o, a_o) \sim \mathcal{D}_O$ to form a training batch $B$
 5:    Compute $\log \pi(a|s)$ values for samples in $B$ using the learned policy $\pi$
 6:    Compute Q-function output values $Q(s, a)$ using sampled $(s, a)$ and $\mathcal{B}^\pi Q(s, a)$ using Eqn. 5
 7:    Update $Q$ by minimizing the learning objective given in Eqn. 4
 8:    **if** t mod $t_{freq}$ **then**
 9:      Update $\pi$ by minimizing the learning objective in Eqn. 6
10:    **end if**
11: **end for**

---

## 5 Experiments

In this section, we analyze the effectiveness of ROIDA for offline IL by utilizing an unlabeled auxiliary dataset. We begin by explaining our experimental setup, including the datasets used and the baseline methods for comparison. Next, we evaluate ROIDA against these baselines across multiple imitation learning scenarios. Specifically, our experiments aim to answer two key questions:

1. How robust is ROIDA when the quality of the auxiliary data varies? (Sec. 5.3.1)

2. How does ROIDA compare to other methods as the size of the expert dataset changes? (Sec. 5.3.2)

In addition to benchmarking against other methods, we perform ablation studies to analyze the contribution of each component of our framework and its scalability.

### 5.1 Experimental setup

We conduct experiments on locomotion and manipulation tasks from the D4RL benchmark (Fu et al., 2020).

**Locomotion**  We use 4 MuJoCo environments for the locomotion tasks: *hopper*, *halfcheetah*, *walker2d*, and *ant*. For expert demonstrations in each environment, we use the corresponding dataset: *hopper-expert-v2*, *halfcheetah-expert-v2*, *walker2d-expert-v2*, and *ant-expert-v2*. For the sub-optimal demonstrations, we source trajectories from the respective *random-v2* datasets. To create $\mathcal{D}_E$, we randomly sample 3, 5, or 7 trajectories per environment depending on the chosen setting. For $\mathcal{D}_O$, we create 3 settings per environment: 1000 randomly sampled sub-optimal trajectories plus another 0, 3, or 5 expert trajectories. This allows us to test our method's ability to identify and leverage expert demonstrations within different mixes of sub-optimal and expert data.

**Manipulation**  We evaluate our method on 4 ADROIT manipulation tasks from using a simulated 24 DoF hand: *pen twirling*, *hammering a nail*, *opening a door*, and *relocating a ball*. For expert demonstrations, we sample 50 trajectories from *pen-expert-v1*, *hammer-expert-v1*, *door-expert-v1*, and *relocate-expert-v1* respectively. For the sub-optimal demonstrations, we use 1000 trajectories from the datasets *pen-cloned-v1*, *hammer-cloned-v1*, *door-cloned-v1*, and *relocate-cloned-v1*, plus 0, 30, or 50 expert trajectories from the corresponding *expert-v1* datasets.

Note that for all the evaluations, we report the average of the mean normalized score for the last 10 evaluations of training over 5 random seeds. Additional implementation details for our method can be

found in Appendix A.2. We also present evaluations using the *rliable* (Agarwal et al., 2021) framework in Appendix A.3.

### 5.2 Baselines

We compare ROIDA against the following algorithms:

- **BC-exp:** BC-exp denotes behavioral cloning solely on the expert dataset $\mathcal{D}_E$. Since $\mathcal{D}_E$ contains only a small number of expert demonstrations, training only on this data can lead to degraded performance at test time due to compounding errors caused by distribution shift.

- **BC-all:** In this case, both $\mathcal{D}_E$ and $\mathcal{D}_O$ are used to learn the policy. Despite the large number of demonstrations, a significant portion of them are random or of low quality. As a result, the learned policy tends to be sub-optimal because of the inclusion of poor demonstrations.

- **DWBC:** DWBC (Xu et al., 2022) treats the auxiliary data as a mixture of expert and sub-optimal data, and utilizes PU learning to train a discriminator for weighted behavioral cloning. They perform a dual learning strategy where they alternately train the discriminator and the policy by taking the output of each model as an input to the other. Due to relying solely on BC, DWBC performs poorly when the number of expert transitions is low.

- **DemoDICE:** DemoDICE (Kim et al., 2022) conducts state-action distribution matching over $\mathcal{D}_E$ and introduces a regularization constraint to ensure the learned policy remains close to the behavior policy of $\mathcal{D}_O$. It shares the same drawbacks as DWBC due to the second term, resulting in a suboptimal policy especially when $\mathcal{D}_O$ contains a large proportion of noisy data.

- **ORIL:** ORIL (Zolna et al., 2020) first learns a reward function and then performs Critic-Regularized Regression (Wang et al., 2020) to learn the policy by enriching the data using different augmentation strategies.

### 5.3 Results

### 5.3.1 Varying the quality of auxiliary data

Table 1 demonstrates how imitation performance changes with auxiliary data of different quality levels. The Setting column denotes the specific quality level; here, $x/y$ indicates using $x$ expert trajectories in $\mathcal{D}_E$ and $y$ expert trajectories in $\mathcal{D}_O$. We evaluate three auxiliary datasets of increasing quality (higher number of expert demonstrations) for each of the environments. To evaluate each method's capacity to extract maximal information from the auxiliary data, regardless of its quality, we present the average performance across all these settings (shown in gray).

Our results show that ROIDA significantly outperforms the baselines, achieving the best performance on 21 out of 24 tested scenarios. Most notably, ROIDA attains the highest *average performance* across all auxiliary datasets for every environment. As expected, the performance generally increases with higher quality auxiliary data. However, ROIDA is able to extract substantially more information from the auxiliary datasets even when those contain lower quality trajectories. This demonstrates the robustness of our approach across diverse settings, without relying on assumptions about the data. In summary, ROIDA consistently outperforms baselines, especially with lower-quality auxiliary data. This highlights ROIDA's effectiveness at leveraging unlabeled data for offline imitation learning.

The poor BC-exp results highlight the challenge of imitation learning with scarce expert data. BC-all uses all available offline data for cloning, without accounting for potentially low-quality policies in the unlabeled data. This often leads to weaker performance than BC-exp.

Although DWBC achieves the second highest average performance, it degrades substantially compared to ROIDA when auxiliary data quality is poor (5/0 and 5/3). This stems from DWBC's inability to extract

Table 1: Imitation learning performance on locomotion (first 4 rows) and manipulation (next 4 rows) tasks from the D4RL benchmark. Results are shown as the number of expert demonstrations in $\mathcal{D}_O$ is increased. The best performing method on each task is highlighted in red and the second best in blue.

| Env. | Setting | Method | | | | | |
|---|---|---|---|---|---|---|---|
| | | BC-exp | BC-all | DemoDICE | ORIL | DWBC | ROIDA |
| Hopper | 5 / 0 | | $2.12 \pm 0.26$ | $38.56 \pm 8.65$ | $3.83 \pm 1.24$ | $72.04 \pm 36.82$ | $84.63 \pm 16.01$ |
| | 5 / 3 | | $2.38 \pm 0.76$ | $51.00 \pm 17.65$ | $15.69 \pm 18.78$ | $74.62 \pm 11.58$ | $86.66 \pm 21.94$ |
| | 5 / 5 | | $2.67 \pm 1.05$ | $68.82 \pm 15.36$ | $17.83 \pm 21.72$ | $80.85 \pm 23.56$ | $88.45 \pm 8.46$ |
| | *Avg.* | $67.15 \pm 16.03$ | $2.39 \pm 0.76$ | $52.80 \pm 14.40$ | $12.45 \pm 16.59$ | $75.84 \pm 26.11$ | $86.58 \pm 16.42$ |
| HalfCheetah | 5 / 0 | | $2.25 \pm 0.00$ | $2.25 \pm 0.00$ | $2.25 \pm 0.00$ | $9.02 \pm 2.88$ | $15.89 \pm 9.60$ |
| | 5 / 3 | | $2.25 \pm 0.00$ | $3.20 \pm 0.21$ | $2.25 \pm 0.00$ | $16.04 \pm 6.40$ | $18.73 \pm 3.67$ |
| | 5 / 5 | | $2.25 \pm 0.00$ | $4.70 \pm 0.13$ | $2.25 \pm 0.00$ | $21.19 \pm 7.53$ | $24.70 \pm 4.95$ |
| | *Avg.* | $8.70 \pm 2.83$ | $2.25 \pm 0.00$ | $3.38 \pm 0.14$ | $2.25 \pm 0.00$ | $15.41 \pm 5.94$ | $19.78 \pm 6.58$ |
| Walker2D | 5 / 0 | | $1.42 \pm 2.28$ | $105.13 \pm 3.35$ | $0.65 \pm 0.06$ | $106.50 \pm 4.09$ | $108.73 \pm 0.28$ |
| | 5 / 3 | | $0.31 \pm 0.06$ | $107.99 \pm 3.52$ | $0.58 \pm 0.07$ | $108.09 \pm 0.37$ | $108.52 \pm 0.21$ |
| | 5 / 5 | | $0.34 \pm 0.16$ | $106.32 \pm 2.44$ | $10.02 \pm 21.00$ | $108.04 \pm 0.43$ | $108.79 \pm 0.09$ |
| | *Avg.* | $103.12 \pm 11.48$ | $0.69 \pm 1.32$ | $106.48 \pm 3.14$ | $3.75 \pm 12.13$ | $107.54 \pm 2.39$ | $108.68 \pm 0.21$ |
| Ant | 5 / 0 | | $31.48 \pm 0.07$ | $49.85 \pm 6.12$ | $38.31 \pm 7.36$ | $61.33 \pm 11.26$ | $65.73 \pm 25.19$ |
| | 5 / 3 | | $31.49 \pm 0.04$ | $51.64 \pm 6.86$ | $38.82 \pm 21.29$ | $73.03 \pm 6.33$ | $77.52 \pm 4.98$ |
| | 5 / 5 | | $31.46 \pm 0.07$ | $46.97 \pm 11.43$ | $37.30 \pm 13.98$ | $72.92 \pm 22.64$ | $76.68 \pm 9.44$ |
| | *Avg.* | $58.78 \pm 2.46$ | $31.48 \pm 0.06$ | $49.49 \pm 8.47$ | $38.14 \pm 15.31$ | $69.10 \pm 15.05$ | $73.31 \pm 15.79$ |
| Pen | 50 / 0 | | $10.12 \pm 16.08$ | $58.68 \pm 14.95$ | $35.79 \pm 18.71$ | $88.52 \pm 16.14$ | $102.55 \pm 8.92$ |
| | 50 / 30 | | $17.32 \pm 17.43$ | $68.03 \pm 7.26$ | $48.47 \pm 17.88$ | $101.45 \pm 7.85$ | $97.39 \pm 6.91$ |
| | 50 / 50 | | $12.32 \pm 16.40$ | $77.46 \pm 30.08$ | $33.95 \pm 12.94$ | $100.00 \pm 16.66$ | $96.41 \pm 8.64$ |
| | *Avg.* | $73.92 \pm 10.76$ | $13.25 \pm 16.65$ | $68.06 \pm 19.84$ | $39.40 \pm 16.70$ | $96.66 \pm 14.14$ | $98.78 \pm 8.21$ |
| Door | 50 / 0 | | $-0.11 \pm 0.05$ | $0.00 \pm 0.00$ | $0.02 \pm 0.01$ | $6.03 \pm 8.21$ | $9.79 \pm 14.66$ |
| | 50 / 30 | | $-0.12 \pm 0.07$ | $0.03 \pm 0.03$ | $0.01 \pm 0.01$ | $9.01 \pm 8.85$ | $7.29 \pm 9.47$ |
| | 50 / 50 | | $-0.08 \pm 0.05$ | $0.24 \pm 0.48$ | $0.01 \pm 0.01$ | $14.89 \pm 18.51$ | $17.70 \pm 16.42$ |
| | *Avg.* | $5.59 \pm 12.37$ | $-0.10 \pm 0.06$ | $0.09 \pm 0.28$ | $0.01 \pm 0.01$ | $9.98 \pm 12.76$ | $11.59 \pm 13.84$ |
| Hammer | 50 / 0 | | $0.27 \pm 0.01$ | $7.55 \pm 9.78$ | $0.30 \pm 0.01$ | $81.82 \pm 18.14$ | $118.33 \pm 18.71$ |
| | 50 / 30 | | $0.25 \pm 0.01$ | $8.28 \pm 9.09$ | $0.29 \pm 0.01$ | $102.75 \pm 29.83$ | $124.71 \pm 4.31$ |
| | 50 / 50 | | $0.26 \pm 0.01$ | $5.52 \pm 4.80$ | $0.29 \pm 0.01$ | $110.45 \pm 20.49$ | $120.91 \pm 5.47$ |
| | *Avg.* | $73.26 \pm 14.89$ | $0.26 \pm 0.01$ | $7.12 \pm 8.19$ | $0.29 \pm 0.01$ | $98.34 \pm 23.37$ | $121.32 \pm 11.53$ |
| Relocate | 50 / 0 | | $-0.04 \pm 0.04$ | $2.25 \pm 0.77$ | $8.44 \pm 9.72$ | $35.12 \pm 13.59$ | $59.74 \pm 19.47$ |
| | 50 / 30 | | $-0.06 \pm 0.07$ | $3.24 \pm 0.96$ | $11.11 \pm 9.47$ | $51.68 \pm 11.05$ | $62.45 \pm 18.54$ |
| | 50 / 50 | | $-0.04 \pm 0.05$ | $2.12 \pm 1.81$ | $13.67 \pm 9.66$ | $62.94 \pm 17.92$ | $71.89 \pm 10.50$ |
| | *Avg.* | $32.97 \pm 19.28$ | $-0.05 \pm 0.05$ | $2.54 \pm 1.26$ | $11.07 \pm 9.62$ | $49.91 \pm 14.47$ | $64.69 \pm 16.67$ |

useful information from poor quality unlabeled demos. In contrast, ROIDA can utilize the auxiliary data through TD learning, despite poor quality. Both DemoDICE and ORIL perform worse in all settings.

### 5.3.2  Varying the size of the expert dataset

In Table 2, we examine how varying the size of the expert dataset affects policy performance. For locomotion tasks, we fix the auxiliary dataset and set the number of expert trajectories in $\mathcal{D}_O$ to 5, while varying the number in $\mathcal{D}_E$ between 3, 5, and 7. For the Adroit tasks, we hold the number of expert trajectories in $\mathcal{D}_O$ at 50 and vary $\mathcal{D}_E$ from 50 to 70. ROIDA achieves comparable performance to DWBC, slightly outperforming in 7 out of 8 environments. As the amount of expert data in $\mathcal{D}_E$ increases, all the methods improve, as expected. In summary, ROIDA maintains competitive performance across varying expert dataset sizes.

Table 2: Imitation learning performance on locomotion tasks as the number of expert demonstrations in $\mathcal{D}_E$ is increased. The best performing method on each task is highlighted in red and the second best in blue.

| Environment | Setting | Method | | | | | |
|---|---|---|---|---|---|---|---|
| | | BC-exp | BC-all | DemoDICE | ORIL | DWBC | ROIDA |
| Hopper | 3 / 5 | $74.84 \pm 18.88$ | $2.44 \pm 1.36$ | $75.75 \pm 28.31$ | $17.11 \pm 25.33$ | $87.00 \pm 14.70$ | $81.53 \pm 10.77$ |
| | 5 / 5 | $67.15 \pm 16.03$ | $2.67 \pm 1.05$ | $68.82 \pm 15.36$ | $17.83 \pm 21.72$ | $80.85 \pm 23.56$ | $88.45 \pm 8.46$ |
| | 7 / 5 | $78.83 \pm 16.52$ | $2.12 \pm 0.34$ | $72.57 \pm 10.67$ | $23.10 \pm 29.66$ | $87.74 \pm 3.93$ | $90.88 \pm 11.32$ |
| | *Avg.* | $73.61 \pm 17.19$ | $2.41 \pm 1.01$ | $72.38 \pm 19.59$ | $19.35 \pm 25.77$ | $85.20 \pm 16.19$ | $86.95 \pm 10.26$ |
| HalfCheetah | 3 / 5 | $5.40 \pm 3.16$ | $2.25 \pm 0.00$ | $2.25 \pm 0.00$ | $2.25 \pm 0.00$ | $7.87 \pm 1.12$ | $8.59 \pm 0.62$ |
| | 5 / 5 | $14.50 \pm 9.88$ | $2.25 \pm 0.00$ | $4.70 \pm 0.13$ | $2.25 \pm 0.00$ | $21.19 \pm 7.53$ | $24.70 \pm 4.95$ |
| | 7 / 5 | $25.75 \pm 9.37$ | $2.25 \pm 0.00$ | $7.84 \pm 4.28$ | $2.25 \pm 0.00$ | $31.79 \pm 5.76$ | $39.19 \pm 2.10$ |
| | *Avg.* | $15.21 \pm 8.07$ | $2.25 \pm 0.00$ | $4.93 \pm 2.47$ | $2.25 \pm 0.00$ | $20.28 \pm 5.51$ | $24.16 \pm 3.12$ |
| Walker2D | 3 / 5 | $94.23 \pm 15.91$ | $0.21 \pm 0.05$ | $106.84 \pm 2.31$ | $6.89 \pm 25.02$ | $106.49 \pm 3.65$ | $106.95 \pm 3.82$ |
| | 5 / 5 | $103.12 \pm 11.48$ | $0.34 \pm 0.16$ | $106.32 \pm 2.44$ | $10.02 \pm 21.00$ | $108.04 \pm 0.43$ | $108.79 \pm 0.09$ |
| | 7 / 5 | $108.14 \pm 0.55$ | $0.31 \pm 0.06$ | $107.64 \pm 5.51$ | $21.59 \pm 17.22$ | $106.76 \pm 3.65$ | $107.71 \pm 0.17$ |
| | *Avg.* | $101.83 \pm 11.33$ | $0.29 \pm 0.10$ | $106.93 \pm 3.72$ | $12.83 \pm 21.32$ | $107.10 \pm 2.99$ | $107.82 \pm 2.21$ |
| Ant | 3 / 5 | $43.83 \pm 26.82$ | $31.51 \pm 0.04$ | $42.09 \pm 15.02$ | $26.88 \pm 13.50$ | $57.52 \pm 12.40$ | $61.43 \pm 9.22$ |
| | 5 / 5 | $58.78 \pm 2.46$ | $31.46 \pm 0.07$ | $46.97 \pm 11.43$ | $37.30 \pm 13.98$ | $76.27 \pm 26.05$ | $76.68 \pm 9.44$ |
| | 7 / 5 | $80.19 \pm 6.22$ | $31.38 \pm 0.16$ | $69.10 \pm 27.95$ | $52.59 \pm 32.42$ | $89.14 \pm 10.18$ | $95.19 \pm 6.00$ |
| | *Avg.* | $60.93 \pm 15.96$ | $31.45 \pm 0.10$ | $52.72 \pm 19.47$ | $38.93 \pm 21.82$ | $74.31 \pm 17.66$ | $77.77 \pm 8.37$ |
| Pen | 50 / 50 | $73.92 \pm 10.76$ | $12.32 \pm 16.40$ | $77.46 \pm 30.08$ | $33.95 \pm 12.94$ | $100.00 \pm 16.66$ | $96.41 \pm 8.64$ |
| | 70 / 50 | $84.62 \pm 22.74$ | $25.29 \pm 20.72$ | $68.65 \pm 16.88$ | $55.96 \pm 18.43$ | $105.25 \pm 17.86$ | $100.19 \pm 11.55$ |
| | *Avg.* | $79.27 \pm 17.79$ | $18.80 \pm 18.69$ | $73.06 \pm 24.39$ | $44.96 \pm 15.92$ | $102.62 \pm 17.27$ | $98.30 \pm 10.20$ |
| Door | 50 / 50 | $5.59 \pm 12.37$ | $-0.08 \pm 0.05$ | $0.24 \pm 0.48$ | $0.01 \pm 0.01$ | $14.89 \pm 18.51$ | $17.70 \pm 16.42$ |
| | 70 / 50 | $8.23 \pm 13.56$ | $-0.12 \pm 0.03$ | $0.03 \pm 0.03$ | $0.02 \pm 0.06$ | $8.27 \pm 8.56$ | $8.13 \pm 7.71$ |
| | *Avg.* | $6.91 \pm 12.98$ | $-0.10 \pm 0.04$ | $0.13 \pm 0.34$ | $0.02 \pm 0.04$ | $11.58 \pm 14.42$ | $12.92 \pm 12.83$ |
| Hammer | 50 / 50 | $73.26 \pm 14.89$ | $0.26 \pm 0.01$ | $5.52 \pm 4.80$ | $0.29 \pm 0.01$ | $110.45 \pm 20.49$ | $120.91 \pm 5.47$ |
| | 70 / 50 | $96.44 \pm 13.43$ | $0.28 \pm 0.01$ | $9.43 \pm 16.42$ | $1.19 \pm 2.92$ | $115.41 \pm 9.47$ | $119.53 \pm 5.58$ |
| | *Avg.* | $84.85 \pm 14.18$ | $0.27 \pm 0.03$ | $7.47 \pm 12.10$ | $0.74 \pm 2.06$ | $112.93 \pm 15.96$ | $120.22 \pm 5.52$ |
| Relocate | 50 / 50 | $32.97 \pm 19.28$ | $-0.04 \pm 0.05$ | $2.12 \pm 1.81$ | $13.67 \pm 9.66$ | $62.94 \pm 17.92$ | $71.89 \pm 10.50$ |
| | 70 / 50 | $60.78 \pm 15.07$ | $0.00 \pm 0.05$ | $4.02 \pm 2.87$ | $18.39 \pm 7.81$ | $73.65 \pm 6.58$ | $74.29 \pm 3.33$ |
| | *Avg.* | $46.88 \pm 17.30$ | $-0.02 \pm 0.05$ | $3.07 \pm 2.40$ | $16.03 \pm 8.78$ | $68.30 \pm 13.50$ | $73.09 \pm 7.79$ |

## 5.4 Ablation study

In this section, we conduct experiments to analyze each component in ROIDA, and also benchmark the performance as the scale of the auxiliary dataset is varied.

*First*, we systematically remove each part of ROIDA and evaluate the resulting performance impact. The ablation results in Table 3 present the performance across three auxiliary dataset settings: 5/0, 5/3, 5/5 (denoting different auxiliary data qualities).

We begin by examining the impact of ablating the reward weighted BC component by setting $\alpha = 0$. Performance drops indicate the importance of filtering the data using a learned reward model. The precise reward assignment enables ROIDA to discern the varying quality of transitions and utilize them accordingly.

Next, we study the impact of the reward function formulation by directly using the discriminator output as the reward. The results demonstrate that our reward formulation based on the DICE methods improves policy performance compared to directly using the discriminator rewards.

We also examine the choice of PU learning (Eqn. 1) over binary classification for reward learning. PU learning accounts for the possibility of optimal transitions within the auxiliary dataset, whereas binary classification risks misclassifying all auxiliary samples as suboptimal, which could degrade performance.

Table 3: Ablation study on each module's contribution to final policy performance on locomotion tasks.

| Environment | Setting | Method | | | |
|---|---|---|---|---|---|
| | | ROIDA | w/o reward-weighted BC | w/o modified reward | w/o PU learning |
| Hopper | 5 / 0 | $84.63 \pm 16.01$ | $41.68 \pm 39.67$ | $78.07 \pm 11.00$ | $84.55 \pm 9.01$ |
| | 5 / 3 | $86.66 \pm 21.94$ | $78.06 \pm 16.17$ | $92.06 \pm 7.10$ | $82.78 \pm 11.49$ |
| | 5 / 5 | $88.45 \pm 8.46$ | $81.51 \pm 15.70$ | $84.73 \pm 11.26$ | $82.23 \pm 11.23$ |
| | *Avg.* | $86.58 \pm 16.42$ | $67.08 \pm 26.34$ | $84.96 \pm 9.97$ | $83.19 \pm 10.64$ |
| HalfCheetah | 5 / 0 | $15.89 \pm 9.60$ | $9.42 \pm 1.48$ | $13.93 \pm 3.31$ | $15.74 \pm 8.63$ |
| | 5 / 3 | $18.73 \pm 3.67$ | $14.93 \pm 8.61$ | $16.35 \pm 4.44$ | $14.53 \pm 3.02$ |
| | 5 / 5 | $24.70 \pm 4.95$ | $18.22 \pm 4.61$ | $22.53 \pm 9.91$ | $16.68 \pm 7.26$ |
| | *Avg.* | $19.78 \pm 6.58$ | $14.19 \pm 5.70$ | $17.61 \pm 6.55$ | $15.65 \pm 6.74$ |
| Walker2D | 5 / 0 | $108.73 \pm 0.28$ | $87.72 \pm 28.91$ | $104.65 \pm 7.79$ | $108.08 \pm 0.48$ |
| | 5 / 3 | $108.52 \pm 0.21$ | $108.11 \pm 0.66$ | $104.77 \pm 6.83$ | $104.78 \pm 3.57$ |
| | 5 / 5 | $108.794 \pm 0.090$ | $107.97 \pm 0.55$ | $105.56 \pm 4.81$ | $105.05 \pm 4.79$ |
| | *Avg.* | $108.68 \pm 0.21$ | $101.27 \pm 16.70$ | $104.99 \pm 6.60$ | $105.97 \pm 3.46$ |
| Ant | 5 / 0 | $65.73 \pm 25.19$ | $57.42 \pm 20.89$ | $49.37 \pm 11.64$ | $70.05 \pm 18.71$ |
| | 5 / 3 | $77.52 \pm 4.98$ | $64.40 \pm 11.52$ | $62.94 \pm 10.27$ | $69.77 \pm 10.31$ |
| | 5 / 5 | $76.68 \pm 9.44$ | $58.95 \pm 17.68$ | $76.14 \pm 18.52$ | $69.08 \pm 11.06$ |
| | *Avg.* | $73.31 \pm 15.79$ | $60.25 \pm 17.14$ | $62.81 \pm 13.95$ | $69.63 \pm 13.89$ |

Table 4: Performance on the Hopper task as the reward threshold $\tau$ is varied.

| Setting | Reward threshold | | |
|---|---|---|---|
| | $\tau = 0$ | $\tau = 1$ (Ours) | $\tau = 2$ |
| 5 / 0 | $83.20 \pm 24.58$ | $84.63 \pm 16.01$ | $70.24 \pm 27.75$ |
| 5 / 3 | $86.82 \pm 6.85$ | $86.66 \pm 21.94$ | $81.91 \pm 18.33$ |
| 5 / 5 | $88.23 \pm 11.42$ | $88.45 \pm 8.46$ | $90.70 \pm 6.88$ |

When the auxiliary dataset contains only suboptimal transitions (5/0), both methods perform similarly. However, as the diversity of the auxiliary dataset increases (5/3 and 5/5), the performance of binary classification decreases. This drop occurs because binary classification incorrectly labels some optimal transitions as suboptimal, negatively impacting overall performance.

*Second*, we study the impact of the reward threshold $\tau$. Our specific reward formulation leads to rewards in the range $[-2.19, 2.19]$. Based on this, we decide to set the threshold $\tau = 1$ to strike a balance between utilizing the "good" transitions in the auxiliary data and neglecting the poor quality transitions. To evaluate the impact of this threshold, we perform a study where we vary the threshold's value using the Hopper environment and report the results in Table 4.

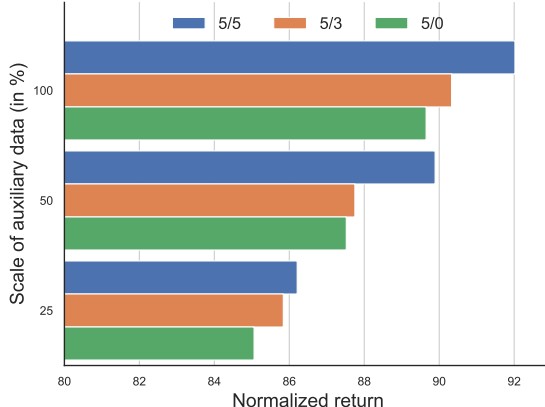

Figure 3: **Scalability to the size of the auxiliary dataset.** We visualize the performance of ROIDA on the Hopper environment as the number of *random* transitions is varied. Here, we show 3 scenarios corresponding to different proportions of the D4RL random set. This highlights ROIDA's ability to learn policies even when the expert and noisy data ratio is quite imbalanced.

We observe that setting $\tau = 0$ does not impact the results too much since it is situated at the midpoint of the reward range. When we increase the threshold to $\tau = 2$ we see a drop in performance in the 5/0 setting; in this setting, no experts are present and thus, the majority of the performance increase should come from the "good" transitions which are rejected due to the high value of the threshold. The performance increase between 5/0 and 5/3 also sheds light on the threshold parameter. When a higher threshold is set, the increase between the settings is quite high, indicating that the performance improvements are coming mostly from the expert data in the auxiliary set and not from the "good" transitions.

*Third*, we change the scale of the auxiliary dataset and analyze the impact on the results. We achieve this by including 25%, 50% or 100% of the entire random set from D4RL (instead of the fixed 1000 trajectories in the previous experiments). As shown in Fig. 3, even with an increased number of suboptimal demonstrations, performance improves as more expert trajectories are added. This clearly highlights ROIDA's ability to distinguish expert data even when the ratio of expert to noisy data is highly skewed. We also observe that adding more auxiliary data while keeping the number of expert trajectories fixed slightly improves performance, indicating ROIDA's ability to extract information from suboptimal data.

# 6 Conclusion

We propose ROIDA, a simple yet effective framework for offline imitation that can maximize utilization of an unlabeled auxiliary dataset of unknown quality alongside a small set of expert demonstrations. Unlike previous methods that make assumptions about auxiliary dataset quality, ROIDA can seamlessly leverage uncurated, unlabeled offline datasets without relying on any quality assumptions. We demonstrate ROIDA's efficacy on multiple manipulation and locomotion tasks, encompassing a wide variety of auxiliary dataset quality settings. The consistent performance gains over baselines validate ROIDA's ability to unlock the full potential of heterogeneous offline datasets without relying on quality assumptions.

**Limitations**

While ROIDA demonstrates strong performance across various environments, we believe there is still room for improvement in the reward estimation process. To investigate this, we conduct an experiment shown in Table 5, where we substitute the estimated reward with the ground-truth reward from the D4RL benchmark. The results indicate a performance gap between the estimated and ground-truth rewards. This finding suggests that our method could potentially achieve higher performance if the reward estimation process is further refined and improved.

Table 5: Performance on the Hopper task with ground-truth rewards.

| Setting | Method | |
|---|---|---|
| | ROIDA | ROIDA w/ GT rewards |
| 5 / 0 | $84.63 \pm 16.01$ | $94.63 \pm 20.76$ |
| 5 / 3 | $86.66 \pm 21.94$ | $98.53 \pm 11.70$ |
| 5 / 5 | $88.45 \pm\ 8.46$ | $104.46 \pm\ 5.42$ |
| *Avg.* | $86.58 \pm 16.42$ | $99.21 \pm 14.11$ |

In our particular framework, the reward estimation can be improved by an accurate choice of the hyperparameter $\eta$ by performing mixture proportion estimation (Zhu et al., 2023; Ramaswamy et al., 2016). However, this is beyond the scope of our work. In order to avoid any assumption about the auxiliary data in our work, we have chosen $\eta = 0.5$ which is an unbiased estimate. We also provide additional results with $\eta = 0.3$ and $\eta = 0.7$ in Table 6. Here, we obtain better results when $\eta$ is closer ($\eta = 0.3$ is closer than $\eta = 0.7$) to the true ratio between expert and suboptimal demonstration in the auxiliary dataset (0.01 for setting 5/0, 0.12 for setting 5/3 and 0.19 for setting 5/5). Since this true ratio is unknown, estimating it would be a problem in its own right, which could then be combined with our method.

Additionally, the current implementation of ROIDA is designed for a single-task setting. An interesting avenue for future work is to extend our framework to multi-task or goal-conditioned settings, where the model

Table 6: Performance on the Hopper task with varying $\eta$.

| Setting | Method | | |
|:---:|:---:|:---:|:---:|
| | $\eta = 0.3$ | $\eta = 0.5$ | $\eta = 0.7$ |
| 5 / 0 | $88.12 \pm 14.93$ | $84.63 \pm 16.01$ | $82.40 \pm 20.76$ |
| 5 / 3 | $90.42 \pm 18.84$ | $86.66 \pm 21.94$ | $84.85 \pm\ 7.13$ |
| 5 / 5 | $91.02 \pm\ 7.97$ | $88.45 \pm\ 8.46$ | $86.45 \pm 18.61$ |
| *Avg.* | $89.85 \pm 14.62$ | $86.58 \pm 16.42$ | $84.57 \pm 14.39$ |

is trained on datasets from different tasks or goals. In such a scenario, ROIDA could be used to learn a new, unseen, but related task based on these multi-task datasets. This extension would open up possibilities for more generalizable and flexible reinforcement learning applications across various tasks.

**Broader impact statement**

Training robots for various tasks using human demonstrations is well-established. However, obtaining high-quality demonstrations in large numbers is extremely challenging and impractical in many cases. Our work provides a method where a robot can learn from a mixture of a limited number of expert high-quality demonstrations and a large number of lower-quality demonstrations. This is a more practically feasible setting and offers promise for developing more efficient approaches to train robots. One possible risk is that the robot can learn unsafe behaviors since the training set may have large numbers of non-expert demonstrations. However, since this is an offline training procedure, the risk is very minimal and can be mitigated through evaluations in lab settings.

**Acknowledgment**

This work was partially supported by The National Science Foundation Award No. 2326309, National Institute for Food and Agriculture Award No. 2021-67022-33453 and the UC Multi-campus Research Programs Initiative.

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

# A Appendix

## A.1 Additional comparisons

We present additional comparisons in Table 7 with another recent unpublished algorithm, BCDP (Shao et al., 2023), which tailors an offline IL algorithm specifically for scenarios where the auxiliary offline dataset contains no expert data. While this design can enhance performance when the auxiliary dataset lacks expert demonstrations, it can lead to suboptimal performance as the proportion of expert data in the auxiliary dataset increases. The results indicate the ROIDA significantly outperforms BCDP on all the seven environments.

Table 7: Imitation learning performance on locomotion (first 3 columns) and manipulation (next 4 columns) tasks from the D4RL benchmark. For each locomotion environment the performance is averaged across scenarios containing different number of expert trajectories in $\mathcal{D}_O$ (5/0, 5/3, 5/5). For each manipulation environment the performance is averaged across scenarios containing different number of expert trajectories in $\mathcal{D}_O$ (50/0, 50/30, 50/50).

| Method | Environment | | | | | | |
|---|---|---|---|---|---|---|---|
| | Hopper | Walker2D | Ant | Pen | Door | Hammer | Relocate |
| BCDP | $65.45 \pm 14.81$ | $103.16 \pm 8.62$ | $47.38 \pm 17.55$ | $11.95 \pm 14.90$ | $3.23 \pm 9.52$ | $0.27 \pm 0.02$ | $0.44 \pm 1.14$ |
| ROIDA | $86.58 \pm 16.42$ | $108.68 \pm 0.21$ | $73.31 \pm 15.79$ | $98.78 \pm 8.21$ | $11.59 \pm 13.84$ | $121.32 \pm 11.53$ | $64.69 \pm 16.67$ |

## A.2 Experimental details

### A.2.1 Implementation

We largely follow the architecture and hyperparameters from DWBC (Xu et al., 2022) for fair comparison. The policy network is a 3-layer MLP with 256 hidden units and tanh outputs. The discriminator is a 4-layer MLP with 128 hidden units, with sigmoid outputs clipped to [0.1, 0.9]. In the PU learning objective 1, we replace the non-differentiable *max* with the *softplus* function to make the loss function differentiable. The Q-function network is an MLP of 3 layers with 256 units. All networks use ReLU activations and the Adam optimizer.

The discriminator learning rate is set to $1e^{-4}$ and a cosine annealing scheduler is added. The policy and Q-function learning rate is set to $3e^{-4}$, with a policy weight decay of 0.005. The balancing factors $\alpha$ and $\beta$ are set dynamically based on the loss ratios. Considering the batch-wise BC loss on expert data to be $\lambda_1$, the batch-wise weighted BC loss on auxiliary data to be $\lambda_2$, and batch wise Q-function loss to be $\lambda_3$, then $\alpha = 0.01 * \frac{\lambda_1}{\lambda_2} * \frac{1}{7.5}$ and $\beta = 0.01 * \frac{\lambda_1}{\lambda_3} * \frac{1}{7.5}$. The additional factor of $\frac{1}{7.5}$ emphasizes the BC loss on expert data, and is adopted from previous work. The discount factor $\gamma$ is 0.5. The frequency of actor model update, $t_{freq}$ is set to 3 for all the environments.

The DICE reward function bounds $\tilde{r}(s, a)$ between [-2.2, 2.2]. For filtering high quality data, we use $\tau = 1$.

All experiments are conducted using PyTorch on a single RTX 3090 GPU.

### A.2.2 Dataset

All datasets are from D4RL (Fu et al., 2020), an offline IL benchmark. Expert trajectories for locomotion tasks are from *<environment>-expert-v2*. Expert trajectories for manipulation tasks are from *<task>-expert-v1*. Sub-optimal transitions are from *<environment>-random-v1* (locomotion) and *<task>-cloned-v1* (manipulation). Table 8 details the number of trajectories and transitions in each dataset. The column Transitions$^\dagger$ refers to the full D4RL datasets for the corresponding random transitions with total number of trajectories given in brackets.

Table 8: Details on total number of trajectories and transitions in the expert and auxiliary datasets for each of the settings in locomotion and manipulation tasks

| Environment | Setting | Expert data $\mathcal{D}_E$ | | Auxiliary Data $\mathcal{D}_O$ | | |
|---|---|---|---|---|---|---|
| | | Trajectories | Transitions | Trajectories | Transitions | Transitions$^\dagger$ |
| Hopper | 5 / 0 | 5 | 5000 | 1000 | 21723 | 999996 (45239) |
| | 5 / 3 | 5 | 5000 | 1003 | 24723 | |
| | 5 / 5 | 5 | 5000 | 1005 | 26723 | |
| | 3 / 5 | 3 | 3000 | 1005 | 26723 | |
| | 7 / 5 | 7 | 7000 | 1005 | 26723 | |
| Halfcheetah | 5 / 0 | 5 | 5000 | 999 | 999999 | 999999 (999) |
| | 5 / 3 | 5 | 5000 | 1002 | 1002999 | |
| | 5 / 5 | 5 | 5000 | 1004 | 1004999 | |
| | 3 / 5 | 3 | 3000 | 1004 | 1004999 | |
| | 7 / 5 | 7 | 7000 | 1004 | 1004999 | |
| Walker2d | 5 / 0 | 5 | 5000 | 1000 | 19877 | 999997 (48907) |
| | 5 / 3 | 5 | 5000 | 1003 | 22877 | |
| | 5 / 5 | 5 | 5000 | 1005 | 24877 | |
| | 3 / 5 | 3 | 3000 | 1005 | 24877 | |
| | 7 / 5 | 7 | 7000 | 1005 | 24877 | |
| Ant | 5 / 0 | 5 | 4465 | 1000 | 180912 | 999930 (5821) |
| | 5 / 3 | 5 | 4465 | 1003 | 183912 | |
| | 5 / 5 | 5 | 4465 | 1005 | 185912 | |
| | 3 / 5 | 3 | 3000 | 1005 | 185377 | |
| | 7 / 5 | 7 | 6465 | 1005 | 185912 | |
| Pen | 50 / 0 | 50 | 5000 | 1000 | 99881 | 499886 (3754) |
| | 50 / 30 | 50 | 5000 | 1030 | 102862 | |
| | 50 / 50 | 50 | 5000 | 1050 | 104862 | |
| | 70 / 50 | 70 | 7000 | 1050 | 104862 | |
| Door | 50 / 0 | 50 | 10000 | 1000 | 200000 | 999939 (4357) |
| | 50 / 30 | 50 | 10000 | 1030 | 206000 | |
| | 50 / 50 | 50 | 10000 | 1050 | 210000 | |
| | 70 / 50 | 70 | 14000 | 1050 | 210000 | |
| Hammer | 50 / 0 | 50 | 10000 | 1000 | 200000 | 999872 (3605) |
| | 50 / 30 | 50 | 10000 | 1030 | 206000 | |
| | 50 / 50 | 50 | 10000 | 1050 | 210000 | |
| | 70 / 50 | 70 | 14000 | 1050 | 210000 | |
| Relocate | 50 / 0 | 50 | 10000 | 1000 | 200000 | 999724 (3747) |
| | 50 / 30 | 50 | 10000 | 1030 | 206000 | |
| | 50 / 50 | 50 | 10000 | 1050 | 210000 | |
| | 70 / 50 | 70 | 14000 | 1050 | 210000 | |

### A.3 RLiable evaluation

In this section, we present evaluations using the *rliable* (Agarwal et al., 2021) framework for our algorithm ROIDA  and DWBC, which is the closest contender. The *rliable* framework aims to reliably evaluate performance with a limited number of runs by employing a rigorous evaluation methodology that accounts for uncertainty in results. It presents more robust and efficient aggregate metrics, such as interquartile mean (IQM) scores, to achieve small uncertainties in the evaluation outcomes.

We group the environments into two benchmarks, *locomotion* and *adroit*. Locomotion contains the Hopper, Walker2D and Ant environments and consists of 9 tasks (3 environments × [5/0, 5/3, 5/5]). Adroit contains the Pen, Door, Hammer and Relocate environments and consists of 12 tasks (4 environments × [50/0, 50/30, 50/50]). We divide the scores by 100 to obtain values in the [0, 1] range, as used in the rliable paper. Using

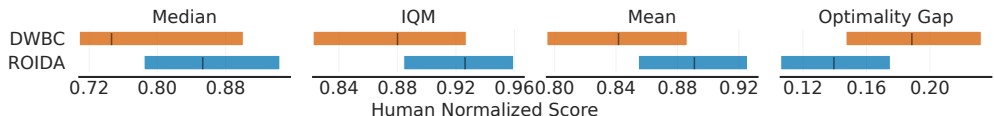

Figure 4: *rliable* evaluation on the Locomotion benchmark.

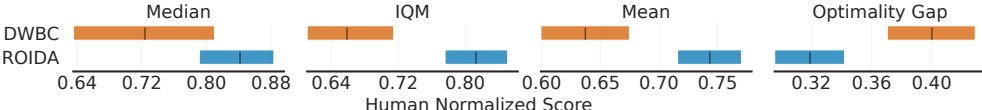

Figure 5: *rliable* evaluation on the Adroit benchmark.

this setup, we report the following performance metrics with 95% confidence intervals: 1) Median performance (higher better), 2) Mean performance (higher better), 3) IQM (higher better), and 4) Optimality gap (lower better).

## A.4 Analysis on hyperparameters

In the main paper, we study the impact of $\eta$ and $\tau$. In this section, we analyze the effect of varying the weighting factors $\alpha$ and $\beta$, which control the contributions of the weighted BC loss and the Q-learning loss, respectively. The results are presented in Figure 6. We vary both hyperparameters across a range from 0.001 to 1.0. Based on the ablation study for the Hopper-5/5 scenario, we select 0.01 for both $\alpha$ and $\beta$ across all setups. These values are then dynamically scaled according to the loss balancing scheme outlined in Section A.2.

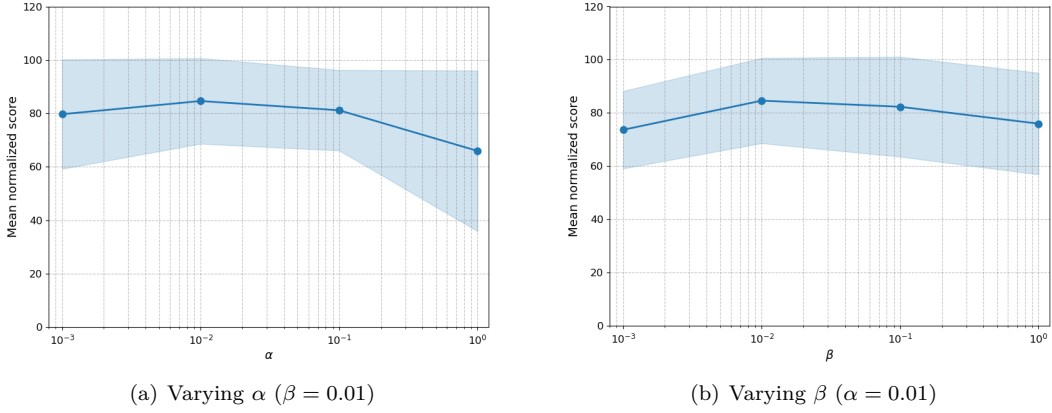

(a) Varying $\alpha$ ($\beta = 0.01$)  (b) Varying $\beta$ ($\alpha = 0.01$)

Figure 6: Performance on Hopper-5/5 as $\alpha$ and $\beta$ are varied.

## A.5 Reward visualization

To understand the accuracy of the proposed reward learning and filtering strategy, we visualize the overlap between the distribution of high-reward transitions as estimated by ROIDA and the overall set of transitions for the Hopper environment. As shown in Figure 7, ROIDA correctly identifies the ground-truth high-reward transitions as *expert* transitions.

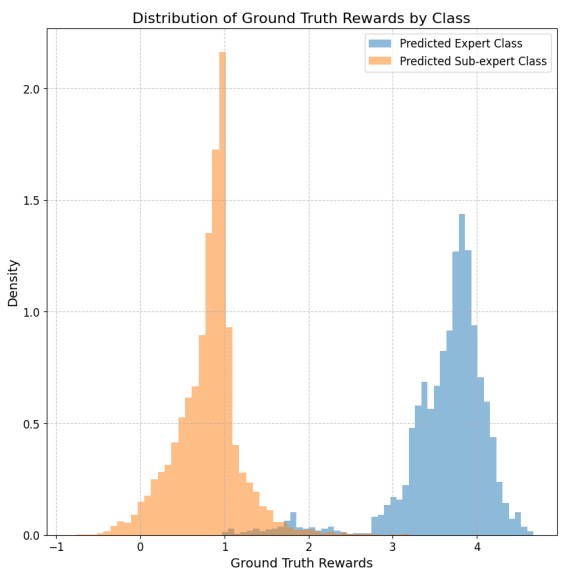

Figure 7: **Reward estimation accuracy on Hopper 5/5.** By combining PU learning-based reward estimation with the filtering strategy, ROIDA effectively identifies the high-reward transitions.

