# OpenReview forum: "Robust Offline Imitation Learning from Diverse Auxiliary Data"
_TMLR — Accepted by TMLR_

### Review · Reviewer_PAuM · 2024-12-03

**Summary Of Contributions:**

This paper introduces ROIDA (Robust Offline Imitation from Diverse Auxiliary Data), a novel offline imitation learning algorithm designed to learn effectively from a combination of limited expert demonstrations and a larger auxiliary dataset of potentially varying quality. Existing methods often struggle with performance fluctuations depending on the quality of the auxiliary data, as they rely on assumptions about the presence of high-reward transitions.  ROIDA addresses this limitation by employing a two-pronged approach: it first identifies high-quality transitions from the entire auxiliary dataset using a learned reward function. These high-reward samples are combined with expert demonstrations for weighted behavioral cloning. For lower-quality samples, it applies temporal difference learning to steer the policy toward high-reward states, improving long-term returns.

Experiments on the D4RL benchmark demonstrate that ROIDA consistently outperforms existing offline imitation learning methods across various environments and auxiliary dataset compositions.  Its robustness to varying data quality stems from its ability to extract information from both high-quality and low-quality transitions, making it a promising approach for practical scenarios where the quality of auxiliary data is unknown or unreliable.  Ablation studies further confirm the contribution of each component of ROIDA to its overall performance.

The key contributions of this paper are:
- Point out the unrealistic assumptions of previous offline IL methods, particularly regarding the composition of the auxiliary set.
- Design an offline IL algorithm, ROIDA, that addresses the limitations of these different methods and remains robust to the quality of demonstrations in the auxiliary dataset.

**Audience:**

Yes

**Claims And Evidence:**

Yes

**Requested Changes:**

See weaknesses.

Additional questions:

- In Figure 1, the caption states "given expert set is kept fixed," yet the y-axis indicates varying numbers of expert trajectories. Could you clarify this inconsistency?
- Section 4.3 mentions, "Instead of directly employing the rewards as weights, we employ a direct thresholding scheme to exclude highly sub-optimal state-action tuples." What advantages does this approach have over directly using rewards as weights?

**Strengths And Weaknesses:**

## Strengths

- The paper tackles a practical challenge of limited expert data in offline imitation learning.  The proposed method leverages readily available, albeit often noisy, auxiliary data, which is a realistic scenario in many robotics applications.

- ROIDA presents a novel combination of PU learning and temporal difference learning within the offline IL framework. This approach allows the algorithm to effectively utilize both high-quality and low-quality transitions in the auxiliary dataset.

- A core strength of ROIDA is its robustness to the quality of the auxiliary data. The algorithm performs consistently well across a spectrum of auxiliary dataset compositions, unlike existing methods that are sensitive to the proportion of expert demonstrations. This robustness is crucial for real-world applications where the quality of auxiliary data may be unpredictable.


## Weaknesses

- This paper claims that ROIDA does not impose any assumptions on the composition of the auxiliary dataset. However, it appears to still implicitly rely on certain assumptions about the auxiliary dataset. As described in Section 4.4, low-quality data is utilized to learn a Q function based on the learned reward, which subsequently directs the policy to maximize the long-term return. However, this process implicitly assumes that the low-quality data has relatively good state coverage. If the state coverage is too narrow, the resulting Q function is likely to be highly unreliable, thereby failing to effectively direct the policy.


- As noted by the authors in the limitations section, the reward estimation process based on PU learning has room for improvement. The performance gap observed when using ground truth rewards indicates that further refinement of the reward estimation could enhance overall performance. Additionally, it would be great to explore methods for illustrating the accuracy of the learned reward. For instance, comparing high-reward transitions with the overall set of transitions and analyzing the overlap between the two distributions could provide meaningful insights.

- ROIDA introduces four additional hyperparameters: $\eta$ in PU, $\alpha$ and $\tau$ in reward-weighted BC, and $\beta$ in TD learning. While the paper provides the chosen values, a more comprehensive analysis of the impact of these hyperparameters and practical guidelines for tuning them would be valuable. Although these hyperparameters may enhance performance, they also introduce an additional layer of complexity.

- This paper primarily compares against a few imitation learning baselines. However, offline RL with sparse rewards could also serve as a potential baseline. Specifically, sparse rewards of 1 could be assigned to expert trajectories and 0 to the auxiliary datasets, followed by running offline RL on this setup.

---

> ### Author Response · Authors · 2024-12-18
>
> **[W1] Auxiliary dataset coverage:**
>
>  We agree that the quality of state coverage is important for learning a reliable Q function. To investigate this, we experiment with auxiliary datasets of different sizes, as shown in Figure 3, where the size of the dataset serves as a proxy for state coverage. While broader state coverage generally improves performance by providing a more comprehensive exploration of the state space, ROIDA remains stable and effective even with auxiliary datasets that have more limited state coverage.
>
> **[W2] Reward estimation visualization:**
>
>  We agree that visualizing the learned rewards would provide valuable insights into the method. To address this, we have included a visualization in the appendix that shows the overlap between the distribution of high-reward transitions and the overall set of transitions for the Hopper environment. We believe this will help clarify the behavior of our approach. As shown in Fig. 7, our method effectively captures the high-reward transitions, with only a small number of low-reward transitions being misclassified as high-reward.
>
> **[W3] Hyperparameter analysis:**
>
> To better understand the impact of *all* the hyperparameters, we provide additional analysis in Appendix A.4, in addition to the ones already present in Tables 4,6.
> The rationale behind the chosen values is as follows:
> 1. $\eta$: To avoid making any assumptions about the auxiliary data, we have selected $\eta = 0.5$, which serves as an unbiased estimate. While this choice works well, the reward estimation could be further improved by selecting $\eta$ more accurately through mixture proportion estimation, as discussed in the Limitations section.
> 2. $\tau$: We set $\tau = 1$ to coincide with approx. 0.75 of our reward range. As shown in Table 4, setting $\tau$ to 0 is also reasonable, as it corresponds to the midpoint of the reward range. However, setting $\tau$ too low or too high negatively impacts performance: a very low value allows too many low-reward transitions to pass through, while a very high value results in the method relying predominantly on the expert data.
> 3. $\alpha$, $\beta$: We set both of these to 0.01 based on the Hopper - 5/5 setting and fix them for all remaining 23 setups.
>
> **[W4] Comparison to offline RL:**
>
>  We assign a reward of 1 to all expert set transitions and 0 to all auxiliary set transitions and run TD3BC[A] as our representative offline RL baseline. Given the well-defined expert set, we adapt the original algorithm by restricting the cloning step to the expert transitions in order to improve performance. As demonstrated in Table C, ROIDA outperforms the offline RL baseline.
>
> *Table C: Comparison with offline RL*
>
> | Environment | Setting | ROIDA | TD3BC
> | :-------- | :-------: | :--------: | -------: |
> | Hopper  | 5/0   | $84.63\pm16.01$| $81.11\pm22.24$ |
> | | 5/3     | $86.66\pm21.94$ | $80.67\pm13.41$ |
> | | 5/5    | $88.45\pm8.46$ | $81.13\pm16.79$ |
> | HalfCheetah  | 5/0   | $15.89\pm9.60$ | $14.63\pm8.89$  |
> | | 5/3 | $18.73\pm3.67$ | $16.23\pm11.17$ |
> | | 5/5 | $24.70\pm4.95$ | $15.18\pm2.79$ |
> | Walker2d | 5/0 | $108.73\pm0.28$ | $107.34\pm1.93$ |
> || 5/3 | $108.52\pm0.21$ | $102.63\pm11.84$ |
> || 5/5 | $108.79\pm0.09$ | $102.47\pm11.73$ |
> | Ant | 5/0 | $65.73\pm25.19$ | $66.85\pm12.31$ |
> || 5/3 | $77.52\pm4.98$ | $67.99\pm12.10$ |
> || 5/5 | $76.68\pm9.44$ | $66.93\pm24.47$ |
>
> **Additional questions**
>
> **Clarification regarding Figure 1:** The y-axis in Figure 1 represents the number of expert/optimal trajectories within the auxiliary set, while the expert set itself remains fixed, as noted in the figure caption. We recognize that the term "expert" may be ambiguous in this context, as it is used to refer both to the expert set and to the expert trajectories within the auxiliary set. To enhance clarity, we have revised the y-axis label accordingly.
>
> **Clarification regarding thresholding:** Since we do not make any assumptions about the quality of the auxiliary dataset, it is possible that some samples in this set are highly suboptimal. To address this, we employ the filtering strategy to ensure that these low-quality samples do not contribute, even minimally, to the cloning objective. Without this filtering, the policy could still be incentivized to consider these transitions for direct imitation via weighted BC, which we seek to avoid.
>
> [A] Fujimoto, Scott, and Shixiang Shane Gu. "A minimalist approach to offline reinforcement learning." Advances in neural information processing systems 34 (2021): 20132-20145.

---

### Review · Reviewer_vPy3 · 2024-12-03

**Summary Of Contributions:**

The work introduces an approach to effectively learn policies from a substantial amount of pre-collected demonstrations not necessarily optimal. The algorithm presents to main stages: (1) a reward model training via Positive-Unlabeled learning and (2) a policy training step which combines together BC, W-BC and TD3. This approach outperforms the presented baselines.

**Audience:**

Yes

**Claims And Evidence:**

Yes

**Requested Changes:**

1. In the problem setting it is claimed that $N_0 >> N_E$, this assumption does not seem to hold true in the proposed experiments or at least I find the notation confusing: what do 0, 3, and 5 stem for when referred to datasets.

2. There is a substantial number of hyperparameters that require at least an ablation study ($\tau$, $\eta$, $\alpha$, $\beta$). I see most of this analysis done in the appendix. I would consider adding it to the main text.

3. Limitations not discussed in the main text. Is the method robust to mismatches between the learning agent and the behavioral agent (the agent from which the data are collected)?

4. Release the code.

5. I find the setting quite interesting since it is an attempt to close the gap between what is assumed in OfflineRL and IL and what happens when dealing with real-world scenarios. I would find interesting to further test this approach in a multi-task or goal-conditioned setting where datasets are collected for different tasks/goals and the approach is used to learn an unseen, but related, task.

**Strengths And Weaknesses:**

## Strengths:
Well-presented and clear, strong empirical results.

## Weaknesses:
Incremental, lack of theoretical analysis for the proposed method, reproducibility (no code available).

---

> ### Author Response · Authors · 2024-12-18
>
> **[C1] Size of expert vs auxiliary datasets:**
>
>  The notation x/y indicates x expert trajectories in the expert set and y expert trajectories plus 1000 suboptimal trajectories in the auxiliary set. Therefore, in all configurations (5/0, 5/3, 5/5), the auxiliary dataset is substantially larger than the expert dataset (with 5 expert trajectories compared to 1000, 1003, and 1005 auxiliary trajectories, respectively). For further clarification, Table 8 in the appendix provides the exact number of transitions for each environment.
>
> **[C2] Ablation studies:**
>
> Following the suggestions of Reviewer PAuM and KxMB, we have conducted additional ablation experiments on reward learning (Table 3) and hyperparameters (Figures 4,6; Appendix A.4). In response to your feedback, we have also moved some of these experiments (Tables 3, 6) to the main text.
>
> **[C3 p1] Limitations:**
>
> The limitations section, previously presented in the appendix, has now been moved to the main text for better visibility.
>
> **[C3 p2] Robustness to mismatches between learning agent and behavioral agent:**
>
>  We do not collect the auxiliary set using our own behavioral agent; instead, we use pre-collected data from the D4RL dataset. As a result, our learning agent does not leverage any inherent similarity between itself and the behavioral agent. Additionally, the average reward values between the expert and auxiliary sets, as shown in Table B, demonstrate a clear mismatch between the two. Despite this, the results in Table 1 show that this mismatch does not negatively affect ROIDA's performance.
>
> *Table B: Mismatch between expert and auxiliary dataset on Hopper*
>
> | | 5/0 | 5/3 | 5/5 |
> |:---|:---:|:---:|---:|
> |Avg. reward in expert set| 3.63 | 3.63 | 3.63 |
> |Avg. reward in auxiliary set| 0.82 | 1.13 | 1.32|
>
> **[C4] Code:**
>
>  The code has been provided as supplementary material for reproducibility.
>
> **[C5] Extension to multi-task settings:**
>
>  Thank you for the insightful suggestion! As you pointed out, ROIDA is currently designed for a single-task setting, where the auxiliary dataset consists of suboptimal demonstrations at various levels of quality for the same task. Extending this approach to a multi-task or goal-conditioned setting would indeed require modifications, such as training task-conditioned reward models for each task, even though the underlying PU learning objective could remain the same. However, dealing with unseen tasks would necessitate additional considerations, such as access to expert demonstrations or other forms of auxiliary data. While this is an exciting direction for future work, it lies outside the scope of the current paper. We have included a discussion of these potential extensions as future work under the Conclusion section and look forward to exploring them in future research.

---

### Review · Reviewer_KxMB · 2024-12-04

**Summary Of Contributions:**

The paper introduces ROIDA, a novel offline imitation learning framework designed to handle auxiliary datasets of mixed-quality demonstrations without making assumptions about their composition. Unlike existing methods which know the quality of the auxiliary demonstrations as a priori, ROIDA identifies expert-like transitions from auxiliary datasets using a discriminator trained with Positive-Unlabeled (PU) learning and performs weighted behavioral cloning. For lower-quality transitions, temporal difference learning is used to guide the policy toward high-reward states.

**Audience:**

Yes

**Broader Impact Concerns:**

None.

**Claims And Evidence:**

No

**Requested Changes:**

1. Could you please add some experiments to verify the function of TD learning with sub-optimal state-action pairs? It is unclear that the performance benefits from this additional term if these state-action pairs deviate from the expert data.
2. If the paper lacks a comparison of different reward model training objectives, such as the binary classifier objective versus PU learning, it would be valuable to include this.

**Strengths And Weaknesses:**

## Strengths
1. **Novel Framework:** ROIDA introduces a robust approach to offline imitation learning by leveraging both high- and low-quality auxiliary data without requiring prior knowledge about data quality. This is a significant improvement over existing methods.
2. **Empirical Performance:** The method consistently outperforms baselines on diverse environments (locomotion and manipulation tasks from D4RL), demonstrating its robustness and general applicability.
## Weaknesses
**Limited Ablation studies:** The ablation studies in the paper claim to validate the contribution of each component of ROIDA. However, some key components in the method still lack clear explanations. For exmaple, does TD learning with sub-optimal state-action pairs really improve the performance? Furthermore, I would be more interesting in the training objective of the reward model, like whether the improvement from the binary classifier objective to the PU learning is significant. The statement "We also study the impact of the reward function formulation by directly using the discriminator output as the reward" is somewhat unclear to me. If the third row in Table 3 addresses this point, please feel free to clarify or correct me.

---

> ### Author Response · Authors · 2024-12-18
>
> **[W, C1] Verify the function of TD learning with suboptimal state-action pairs:**
>
> In our experiments, we simulate the scenario where the auxiliary data consists entirely of sub-optimal state-action pairs, using the 5/0 setup for locomotion and the 50/0 setup for manipulation. As shown in Table 1, ROIDA consistently outperforms DWBC, a state-of-the-art weighted BC method, across all environments. This demonstrates the advantage offered by TD learning with suboptimal data, which reintroduces rich state information into the policy training process, thereby enhancing robustness. Since the optimal action is unclear in the auxiliary data (due to suboptimal state-action pairs), the Q-function helps guide the policy toward actions that lead to higher-quality states and subsequently better rewards. This allows for more effective imitation of expert behavior from that state onward.
>
> **[W, C2] Using binary classifier instead of PU learning for reward estimation:**
>
> Thank you for the suggestion. Our choice of PU learning over binary classification for reward learning accounts for the possibility of optimal transitions within the auxiliary data. In contrast, binary classification risks misclassifying all auxiliary data samples as negative or suboptimal, potentially degrading performance.
>
> To highlight this difference, we compare the performance of binary classification versus PU learning in Table A. When the auxiliary data consists entirely of suboptimal transitions (5/0), both methods perform similarly. However, as the auxiliary set becomes more diverse (5/3 and 5/5), the performance of binary classification declines. This drop is due to binary classification incorrectly labeling optimal transitions as suboptimal, which negatively impacts the overall performance.
>
> *Table A: PU learning vs Binary classification*
> | Environment | Setting | ROIDA | ROIDA w/ binary classifier
> | :-------- | :-------: | :--------: | -------: |
> | Hopper  | 5/0   | $84.63\pm16.01$| $84.55\pm9.01$ |
> | | 5/3     | $86.66\pm21.94$ | $82.78\pm11.49$ |
> | | 5/5    | $88.45\pm8.46$ | $82.23\pm11.23$ |
> | HalfCheetah  | 5/0   | $15.89\pm9.60$ | $15.74\pm8.63$  |
> | | 5/3 | $18.73\pm3.67$ | $14.53\pm3.02$ |
> | | 5/5 | $24.70\pm4.95$ | $16.68\pm7.26$ |
> | Walker2d | 5/0 | $108.73\pm0.28$ | $108.08\pm0.48$ |
> || 5/3 | $108.52\pm0.21$ | $104.78\pm3.57$ |
> || 5/5 | $108.79\pm0.09$ | $105.05\pm4.79$ |
> | Ant | 5/0 | $65.73\pm25.19$ | $70.05\pm18.71$ |
> || 5/3 | $77.52\pm4.98$ | $69.77\pm10.31$ |
> || 5/5 | $76.68\pm9.44$ | $69.08\pm11.06$ |
>
> We have added this ablation to the main paper in Table 3.
>
> **[W] Impact of reward formulation:**
>
> You are correct in your assessment. The third row in Table 3 examines the effect of using the reward formulation in Eq. 2 compared to directly utilizing the discriminator output. Note that Table 3 has been updated in the revised paper, the ablation for the reward formulation is now the second from last column.

---

### Author Response · Authors · 2024-12-18

We thank the editors and reviewers for their time on our paper and the suggestions and comments made. We have addressed all the comments in the main paper. A summary of the changes is also listed below.

List of changes:

1. Added new ablation on reward learning
2. Added code as supplementary material
3. Changed layout of paper to include more material in the main paper from the appendix
4. Added new ablation on hyperparameters in appendix
5. Updated figure 1 to improve clarity

All changes have been marked in blue.

---

### Decision · Action_Editor_hmSU · 2025-04-17

**Recommendation:** Accept as is

**Comment:**

The paper meets both acceptance criteria and all reviewers recommended acceptance. The paper was already revised during the review process.

**Audience:**

The problem setting is of practical relevance and in some regards closer to the Offline-RL setting as the more typical Offline-IL setting, which only considers an expert dataset. The proposed approach is reasonable, the empirical results are good, and all reviewers stated that the work is relevant for TMLR's audience.

**Claims And Evidence:**

__Brief summary:__
The submission presents a new methods for offline imitation learning assuming that two datasets are available: an expert dataset and an unlabeled dataset. A discriminator / reward is trained using a loss for positive-unlabeled learning to account for the fact that the unlabeled dataset may include good transitions. The policy is trained using a combination of two losses: 1) weighted BC, where transitions below a given threshold are fully discarded, 2) maximizing the expected Q-Value using all data, where the Q-function is estimated by minimizing the TD-Error. The method is evaluated in simulation (MuJoCo and D4RL), where it achieves strong empirical performance across different compositions of the unlabeled dataset.

__Claims and Evidence:__ The main claims of the submission are with respect to its performance for different dataset compositions. These claims were substantiated by the experiments.